# Trends in Simple and Complex Appendicitis in Children and the Potential Correlation to Common Viral Pathogens—A Retrospective Cohort Study between 2010 and 2019 in The Netherlands

**DOI:** 10.3390/children10121912

**Published:** 2023-12-11

**Authors:** Sarah-May M. L. The, Paul van Amstel, Sophie M. Noordzij, Roel Bakx, Taco. S. Bijlsma, Joep. P. M. Derikx, L. W. Ernest van Heurn, Martijn van der Kuip, Ramon R. Gorter

**Affiliations:** 1Department of Pediatric Surgery, Emma Children’s Hospital, Amsterdam UMC, University of Amsterdam & Vrije Universiteit Amsterdam, 1105 AZ Amsterdam, The Netherlands; s.the@amsterdamumc.nl (S.-M.M.L.T.); p.vanamstel@amsterdamumc.nl (P.v.A.); sophie.noordzij@hilversum.psyq.nl (S.M.N.); r.bakx@amsterdamumc.nl (R.B.); j.derikx@amsterdamumc.nl (J.P.M.D.); e.vanheurn@amsterdamumc.nl (L.W.E.v.H.); 2Amsterdam Reproduction and Development Research Institute, 1081 HV Amsterdam, The Netherlands; 3Department of Surgery, Northwest Hospital, 1815 JD Alkmaar, The Netherlands; t.s.bijlsma@nwz.nl; 4Amsterdam Gastroenterology and Metabolism Research Institute, 1081 HV Amsterdam, The Netherlands; 5Department of Pediatric Infectious Diseases and Immunology, Emma Children’s Hospital, Amsterdam UMC, Amsterdam Infection & Immunity Institute, Vrije Universiteit Amsterdam, 1081 HV Amsterdam, The Netherlands; m.vanderkuip@amsterdamumc.nl

**Keywords:** appendicitis, simple, complex, children, viral pathogens

## Abstract

The aim of this study was to evaluate the annual, seasonal and monthly trends in children with simple and complex appendicitis and their correlation to common viral pathogens in the Netherlands. A consecutive multicenter retrospective cohort study was performed between 2010 and 2019 including children (<18 years) surgically treated for appendicitis. The primary outcome was the distribution of children with simple and complex appendicitis per year, season and month. Relevant seasonal variation was defined as ≥5%. The secondary outcome was a positive correlation of the number of patients with simple and complex appendicitis to common viral pathogens (data anonymously provided by the Dutch Working Group on Clinical Virology from the Dutch Society for Clinical Microbiology (NVMM)). In total, 896 patients were included: N = 524 (58%) patients with simple and N = 372 (42%) with complex appendicitis. Of the children aged 0–5 years, 81% had complex appendicitis, versus 38% in 6–18 years (*p* < 0.001). An overall decline was demonstrated for both simple and complex appendicitis between 2010 and 2019. No seasonal variation was found for simple appendicitis. For complex appendicitis, the highest number of patients was found in spring, and lowest in summer (N = 372, spring 28.2 ± 5.1% versus summer 21.0 ± 5.8%, *p* = 0.011), but the variance was regarded as not relevant (<5% from baseline). A positive correlation was found between complex appendicitis with Adenovirus 40.41 (R = 0.356, 95%CI 0.045–0.604, *p* = 0.026) and simple appendicitis with Adenovirus NON 40.41 (R = 0.332, 95%CI 0.019–0.586, *p* = 0.039), but these correlations did not remain significant after a Bonferroni correction (*p* < 0.003). In conclusion, we found no relevant seasonal variation for simple or complex appendicitis, nor positive correlation with common viral pathogens.

## 1. Introduction

Appendicitis was long thought to be an irreversible disease caused by intraluminal obstruction (for example, by a fecalith or lymphoid hyperplasia). However, the review by Carr demonstrates that the pathogenesis of appendicitis is multifactorial and that in the majority of cases intraluminal obstruction is unlikely to play a role [1]. Factors such as diet and bacterial or viral pathogens have been mentioned, but the exact underlying pathogenesis needs further unravelment. In addition, nowadays, two types of appendicitis are identified: simple, which includes phlegmonous (or non-perforated) appendicitis, and complex appendicitis, which includes gangrenous and perforated appendicitis, with or without abscess formation [2]. Differences between both types are found in terms of patient characteristics, clinical presentation, immunological responses and microbial composition, and optimal treatment strategies [2,3,4,5,6,7]. Unfortunately, diagnosing appendicitis in children in general, but specifically differentiating between both types, remains a challenging endeavor. Efforts have been made recently to improve this by using (non-invasive) biomarkers and development of scoring models [8,9,10,11]. However, the diagnostic accuracy remains insufficient. The evaluation of predisposing factors, epidemiological data of both types and correlating this to potential viral pathogens may enhance knowledge of the underlying pathogenesis and provide handholds for improved differentiation.

Several studies previously evaluated the seasonal variation of appendicitis and the majority of them identified a predominance in summer [12]. The mechanism is not fully understood, but factors such as weather conditions, dietary changes, air pollution, and outbreaks of common viral pathogens have all been suggested as potential explanations for this seasonal variation of appendicitis [12,13,14]. Most findings on seasonal variation are, however, derived from adult and mixed populations. Findings in the pediatric population are rare, show little variation between seasons and/or outcomes seem to vary [15,16,17,18,19,20]. Furthermore, most studies focus on appendicitis in general, with only little data on the difference between simple and complex appendicitis. Interestingly, the overall incidence of appendicitis in children appears to be declining over the years and evaluating differences between simple and complex appendicitis may be worthwhile as Alder et al. found a potential cointegration between influenza and the annual decline of non-perforated (simple), but not perforated (complex), appendicitis [15,21,22,23]. Further data on the trends in the two types of appendicitis and their potential correlation to viral pathogens is unfortunately rare. 

Taken together, the primary aim of this study was to evaluate the annual, seasonal and monthly trends in simple and complex appendicitis in the pediatric population in the Netherlands. The secondary aim was to evaluate whether the number of children with simple and complex appendicitis is correlated to common viral pathogens in the Netherlands. 

## 2. Materials and Methods

Data collection for this study was part of a consecutive multicenter retrospective cohort study in a secondary center (Northwest Hospital (NWZ), Alkmaar, the Netherlands) and a tertiary referral center (Emma Children’s Hospital, Amsterdam University Medical Center (UMC), former location Academic Medical Center (AMC), Amsterdam, the Netherlands) between 1st of January 2010 and 31th of December 2019. The medical ethics committee of the Amsterdam UMC reviewed the study and confirmed that the Dutch Medical Research Involving Human Subjects Act did not apply and the need for full ethical review was therefore waived. This was confirmed by the medical ethics review committee of the NWZ. All children aged <18 years that underwent an appendectomy for acute appendicitis in this period were eligible for inclusion. Children were excluded from data analysis in case they (i) were treated conservatively (due to the lack of histopathological confirmation leading to the inability of accurate classification into either simple or complex appendicitis); (ii) in case of other diagnoses than acute appendicitis (such as malignancy of the appendix, inflammatory bowel disease or chronic appendicitis) or lack of confirmation of appendicitis (non-inflamed appendix encountered during surgery or at histopathological examination); and (iii) in case patients were referred to, or from one of the above mentioned hospitals (due to the risk of missing data). In the tertiary referral center, identified patients (and parents) received written information about the study and the right to object to the use of data. The secondary center waived this procedure due to the extent of the number of patients.

### 2.1. Clinical Data Collection

All data, including baseline characteristics, medical history, physical examination, biochemical and urine analysis, radiological examination and perioperative course, were obtained according to a predefined data form in an online database (Castor Electronic Data Capture). Variables of interest for this particular study were hospital of presentation, age of patients (years), biological sex, date and time of presentation, date and time of surgery, and surgical and histopathological findings (for classification of types of appendicitis as described hereafter). 

### 2.2. Clinical and Pathological Classification

All patients with histopathological confirmation of acute appendicitis were classified into two groups upon their perioperative and histopathological findings in line with the modified classification of Bhangu et al.: simple appendicitis, i.e., phlegmonous appendicitis with or without presence of pus formation and without signs of complex appendicitis; and complex appendicitis, i.e., gangrenous (presence of necrosis) or perforated (a macroscopic visible hole of the appendix or microscopic perforation) appendicitis with or without abscess formation [2]. Separate data will be provided for the sub-group of patients with perforated appendicitis. Classification of patients was performed by two independent authors (S.T. and P.A.). Discrepancies were resolved by plenary discussion or consultation of a third author (R.G.).

### 2.3. Viral Data Collection

To correlate between simple and complex appendicitis with several common viruses in the Dutch population, viral reports were provided by the Dutch Working Group on Clinical Virology from the Dutch Society for Clinical Microbiology (NVMM) and participating laboratories. Data were provided as number of positive tests per virus per week from the first week of January 2010 up to the last week of December 2019. Incidence calculations of positive tests were not provided and tests were obtained from both adults and children. Ethics approval was not applicable as all data was anonymously received. Viruses of interest were: Adenovirus not type 40 and 41 (NON 40.41), Adenovirus type 40 and 41 (40.41), Enterovirus, Influenza type A, Norovirus, Para-influenza type 1, Para-influenza type 2, Para-influenza type 3, Para-influenza type 4, Para-influenza non-typed, Parechovirus, Parvovirus, Rhinovirus, Rotavirus, RS-virus. 

### 2.4. Outcomes

The primary outcome of this study was the distribution of children with simple and complex appendicitis per year, season and month. Categorization into seasons was provided according to the Northern hemisphere: spring consisting of March, April and May; summer of June, July and August; autumn of September, October and November; winter of December, January and February. The secondary outcome of this study was the number of patients with simple and complex appendicitis with a positive correlation to previously mentioned viruses. As data of viral reports were provided per week, we categorized seasons for the secondary outcome according to week numbers (both for viral reports and patients with either simple or complex appendicitis): spring consisted of week 10–22, summer of week 23–35, autumn of week 36–48 and winter of week 49–9. 

### 2.5. Statistics

Fisher’s exact test was used for dichotomous variables and *t*-test or Mann–Whitney U test were used for continuous variables when appropriate. Normality was evaluated with Kolmogorov–Smirnov and Shapiro–Wilk tests of normality. A linear regression analysis was performed to evaluate the annual number of children with simple and complex appendicitis. The distribution per season was calculated in absolute numbers and percentage of patients per season. Relevant seasonal variation was defined as an increase in percentage of at least 5% (or even more preferably, 10%) compared to expected baseline of 25% per season, i.e., a total of less than 20% or more than 30% per season. Correlation between number of patients with simple and complex appendicitis with viruses was analyzed with Pearson correlation analyses. Statistics and graphical representations were performed using IBM SPSS Statistics (Version 28.0) and Graphpad Prism (Version 9). Statistical significance was determined as two-tailed *p* < 0.05. Regarding the correlation between viruses and simple and complex appendicitis, a Bonferroni correction was provided to correct for multiple testing. 

## 3. Results

In total, 987 children were consecutively identified in both participating hospitals in the Netherlands between 1st of January 2010 up to 31th of December 2019. In 938 children, (immediate) surgical treatment was performed for presumed acute appendicitis. The remaining 49 children were (initially) conservatively treated and excluded from analysis. Of the 938 surgically treated patients, the appendix was not resected in four patients after encountering an appendix sana; of the remaining 934 patients, the appendix was resected and sent for histopathological evaluation. Of these, 38 patients were excluded upon predefined criteria (non-inflamed appendix N = 29, 3%; chronic appendicitis N = 5, 0.5%; and malignancy of the appendix N = 4, 0.4%). Subsequently, 896 patients were included for the final analysis (Figure 1): N = 524 (58%) patients with simple appendicitis and N = 372 (42%) with complex appendicitis. In 267 patients (30%), a macroscopic or microscopic perforation was reported during either surgery and/or histopathologic review. An overview of the baseline characteristics of the included patients is provided in Table 1. Differences between children with simple and complex appendicitis were found regarding the hospital of presentation, age at presentation, days of abdominal pain, vomiting (but not nausea), diarrhea, obstipation, urinary tract symptoms, temperature at presentation, leukocyte count, C-reactive protein (CRP) levels, and the presence of a fecalith at imaging.

### 3.1. The Clinical Presentation of Simple and Complex Appendicitis According to Age 

As provided in Table 1 and Figure 2 and Figure 3, an association was found between age and severity of appendicitis with a predominance of complex and specifically perforated appendicitis among children aged 0–5 years compared to children aged 6–18 years. Complex appendicitis was reported in 59/73, 81% of the children aged 0–5 years, versus 313/823, 38% of the children 6–18 years (*p* < 0.001). Perforated appendicitis was reported in 56/73, 77% of the children aged 0–5 years versus 211/823, 26% of the children aged 6–18 years (*p* < 0.001).

### 3.2. The Annual Distribution of Simple and Complex Appendicitis in Children over the Last Decade (2010–2019) 

The average number of patients was 90 per year: a total of 52 for simple appendicitis, and 37 for complex appendicitis, respectively. A decline was found over the years for both children with simple (Y = −3.188X + 67; *p* = 0.001) and complex appendicitis (Y = −2.509X + 48; *p* = 0.033) (Figure 4). The average annual decrease between 2010 and 2019 was 5.0% for simple (total of −37% in 2019 compared to 2010) and 4.6% for complex appendicitis (a total of −34% in 2019 compared to 2010). For children with perforated appendicitis, the decrease was not significant (Y = −0.842X + 30; *p* = 0.277).

### 3.3. The Seasonal and Monthly Distribution of Simple and Complex Appendicitis in Children

Table 2 demonstrates the number of patients with simple and complex appendicitis per season. For simple appendicitis, no significant or relevant variation was found regarding distribution per season. For complex appendicitis, the highest percentage of patients per year was presented in spring, and the lowest in summer: spring 28.2 ± 5.1% versus summer 21.0 ± 5.8%, *p* = 0.011. Although statistically significant, the difference was regarded as not relevant (<5% variation from expected baseline).

In Table 3, the distribution of patients with simple and complex appendicitis is demonstrated per month. The highest percentages of patients with simple appendicitis were found in June (N = 52, 9.9%) and September (N = 53, 10.1%), and the lowest percentage of patients in July (N = 35, 6.7%). The highest percentages of patients with complex appendicitis were found in May (N = 42, 11.3%) and October (N = 38, 10.2%), and the lowest percentage of patients in July (N = 21, 5.6%). 

### 3.4. Do Viral Infections Coincide with Simple and Complex Appendicitis in Children?

A positive correlation was found between Adenovirus 40.41 with complex appendicitis (R = 0.356, 95%CI 0.045–0.604, *p* = 0.026) and Adenovirus NON 40.41 with simple appendicitis (R = 0.332, 95%CI 0.019–0.586, *p* = 0.039 (Figure 5). These correlations were not significant after a Bonferroni correction (0.05/15, = *p* < 0.003). No positive significant correlations were found for Enterovirus, Influenza type A, Norovirus, Para-influenza type 1–4 and non-typed, Parechovirus, Parvovirus, Rhinovirus, Rotavirus, or RS-virus.

## 4. Discussion

In this consecutive multicenter retrospective cohort study, we evaluated the trends in simple and complex appendicitis in children in the Netherlands between 2010 and 2019. We found a clear correlation between age and severity of appendicitis, as complex appendicitis was predominant in children aged 0–5 years. In addition, the overall number of children with both simple and complex appendicitis declined between 2010 and 2019. Regarding seasonal variation, the highest number of patients with complex appendicitis was in spring and the lowest was in summer, but the variation was regarded as not relevant. For simple appendicitis, neither a significant nor relevant seasonal variation was found. A positive correlation was found between Adenovirus 40.41 with complex appendicitis and Adenovirus NON 40.41 with simple appendicitis, but these correlations did not remain significant after a Bonferroni correction for multiple testing.

### 4.1. The Association between the Age of Children and Complex Appendicitis

As previously stated, appendicitis can be divided into two types, simple and complex appendicitis. Differences have been demonstrated in terms of clinical presentation, biochemical results, microbial composition of the appendix and an increased pro-inflammatory response in complex as opposed to simple appendicitis [2,3,4,5,6,7]. Additionally, in this cohort, we provide a clear association between age and severity of appendicitis, as 81% of children aged 0–5 years had complex appendicitis compared to 38% of children aged 6–17 years. More specifically 77% of children aged 0–5 years had perforated appendicitis compared to 26% of children aged 6–17 years. This increased perforation rate in younger children is in line with the findings of Pogorelić et al., who previously evaluated children < 5 years of age with appendicitis and found a positive correlation between perforation rate and age [24]. It should be questioned why these younger children often present with increased disease severity or rather appear to be more susceptible to developing complex (and specifically perforated) appendicitis. It is hypothesized that this could be due to a delay in diagnosis due to lack of awareness of appendicitis as a diagnosis in younger children with often non-specific symptoms. However, Athena’s review by Raveenthiran et al. showed that neonates with perforated (complex) appendicitis as opposed to non-perforated (simple) appendicitis were treated earlier and had lower mortality rates [25]. Other explanations are thus likely to play a role as well. Commonly mentioned factors are the thin appendiceal wall, the lack of protective omentum and potentially the underdeveloped immune system, but the exact mechanism is presumably multi-factorial and largely unknown [26,27,28,29]. To evaluate this gap in knowledge and further unravel the underlying pathogenesis of simple and complex appendicitis, we believe that it may be valuable to specifically look into the differences of simple and complex appendicitis in children aged 0–5 years.

### 4.2. What Causes the Decline in Children with Simple and Complex Appendicitis?

In this cohort in the Netherlands, we found a decline in both children with simple and complex appendicitis between 2010 and 2019. The findings of this study are in line with the previous literature, and suggest that the decline is ongoing. Almstrom et al. demonstrated a total decline of 43.7% between 1987 and 2013 in Sweden, Omling et al. an annual decline of 3.2% between 2001 and 2014 in Sweden, and Rautava et al. an annual decline of 3.3% between 2004 and 2014 in Finland (all pediatric populations) [15,22,23]. We hypothesize that the decline in incidence may be due to a true decline in children with appendicitis, advances in diagnostic work-up, or increased implementation of conservative treatment strategies. 

We opted to exclude data from 2020 and onwards in our cohort study as the COVID-19 pandemic might have influenced the trends in appendicitis in children. Interestingly, some studies have been performed during the COVID-19 pandemic and show a disintegration between simple and complex appendicitis. Ceresoli et al. demonstrated a decrease in non-complicated (simple) appendicitis and negative appendectomy rate, but not of complicated (complex) appendicitis during the COVID-19 pandemic [30]. Van Amstel et al. reported a relative increase in complex appendicitis in children during the (start of the) COVID-19 pandemic due to an absolute decrease in simple appendicitis [31]. In addition, Pogorelić et al. performed a meta-analysis and found an overall higher incidence of complex appendicitis in children during the COVID-19 period [32]. These findings may be a result of a lower referral rate, a more awaiting (out of hospital) approach of patients with appendicitis, and a higher incidence of a nonoperative treatment for (clinically simple) appendicitis. But, more importantly, these results support the presence of two types of appendicitis.

### 4.3. Data from This Cohort Study Show No Relevant Seasonal Variation in Simple or Complex Appendicitis in Children

The seasonal variation of appendicitis has been previously evaluated. Fares et al. demonstrated that the majority of these studies (both adults/mixed populations and children combined) found a peak during summer [12]. Several factors have been suggested as an explanation for the potential seasonal variation such as weather conditions, air pollution, allergens and dietary changes [12,13,14]. For example, a relationship between the incidence of appendicitis and high environmental temperature has been described in some studies [14,16].

Nevertheless, conclusions on seasonal variation should be made after careful consideration as both methodology and definitions of seasons vary between studies. More importantly, looking at the pediatric population only, the results seem to be inconclusive as either the differences between seasons are small, multiple peaks are described, or the highest number of patients is found in winter as opposed to summer (Appendix A) [15,16,17,18,19,20]. Interestingly, Deng et al. (in children only) and Lee et al. (in a mixed population) provided additional insights on the severity of appendicitis and seasonal variation [17,33]. Both reported an overall peak of appendicitis in Summer, but perforated appendicitis was paradoxically higher in fall/winter in the study of Deng et al. and in winter in the study of Lee et al. In contrast to the previous literature, we found no seasonal variation for children with simple appendicitis, but seasonal variation was found for children with complex appendicitis, with the highest number of children in spring (and not summer). The variation in this study (and that of Deng et al.) was however, less than 5% of the expected baseline (25%) and was therefore considered not relevant.

### 4.4. Is There a Role for Viruses in Simple and Complex Appendicitis in Children?

In this study, we found a positive, but weak, correlation between Adenovirus 40.41 with complex appendicitis and Adenovirus NON 40.41 with simple appendicitis. Unfortunately, this correlation did not remain significant after a Bonferroni correction for multiple testing. Interestingly, several viruses, including Adenovirus, have been previously associated with appendicitis when tested in a patient with appendicitis [34,35,36]. In conclusion, we cannot confirm the role of Adenovirus in appendicitis based upon the results of our study, but we hypothesize that Adenoviruses may potentially play a role in the pathogenesis of simple and complex appendicitis. In this light, we hypothesize that Adenovirus may enable an increased pro-inflammatory response in complex appendicitis due to a concurring difference in the microbial composition (as the latter has been previously demonstrated) [3,6]. This theory is supported by the findings of Liu et al., who demonstrated an increased pro-inflammatory response in children <1 years of age with pneumoniae when both viral and bacterial pathogens were present as opposed to viral or bacterial pathogens only [37].

### 4.5. Limitations

This study was limited by the fact that viral data was only available from twenty participating laboratories in the Netherlands, was obtained in both children and adults, and lacked the potential of incidence calculations and data on the clinical severity of viral infections. As data on viral pathogens were not obtained in the cohort of children with appendicitis, a direct link between both types of appendicitis and viral pathogens could not be provided. This should be kept in mind when interpreting the results of the current study. In addition, the sample size of our study population was relatively small (N = 896). Consequently, statistics have only been provided on seasonal variation and not on monthly or weekly variation, and the results could not be stratified for potential confounding variables (such as environmental temperature, diet and patient characteristics). Neither were results stratified for geographic distribution, as both hospitals are within the same region. In addition, the exclusion of non-operatively treated patients and patients lacking histopathological confirmation of appendicitis may have introduced a selection bias. A comparison of patient characteristics between excluded and included patients was not performed. 

### 4.6. Future Studies

Differentiating between simple and complex appendicitis remains challenging but relevant, as treatment strategies may differ. Based upon the findings of this study, it appears that future research on the seasonal variation of both types will not aid in improved (clinical) differentiation. However, it may be worthwhile to specifically look into the differences between children aged 0–5 years with simple and complex appendicitis (and compare this with older children). And, we strongly advise combining this with the testing of viral pathogens in the same patient, especially with Adenovirus in a prospective cohort study. 

## 5. Conclusions

This retrospective cohort study in the Netherlands demonstrated an association between the age of children and complex appendicitis, and a decline in both children with simple and complex appendicitis between 2010 and 2019. No relevant seasonal variation in simple or complex appendicitis, nor positive correlation with common viral pathogens after Bonferroni corrections, were found.

## Figures and Tables

**Figure 1 children-10-01912-f001:**
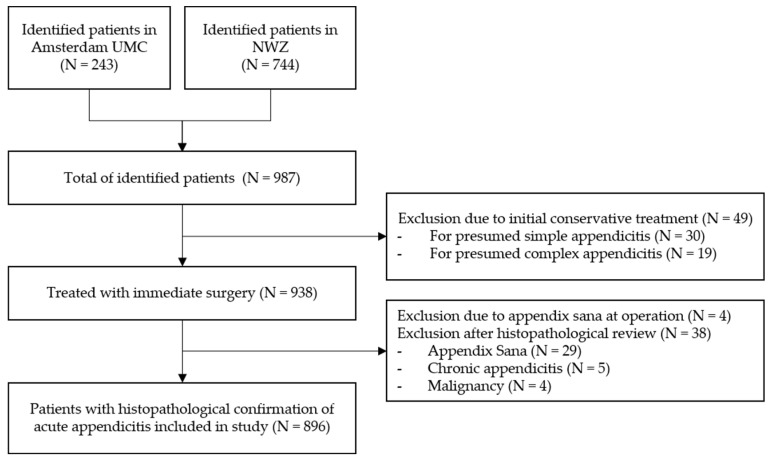
Inclusion Flowchart.

**Figure 2 children-10-01912-f002:**
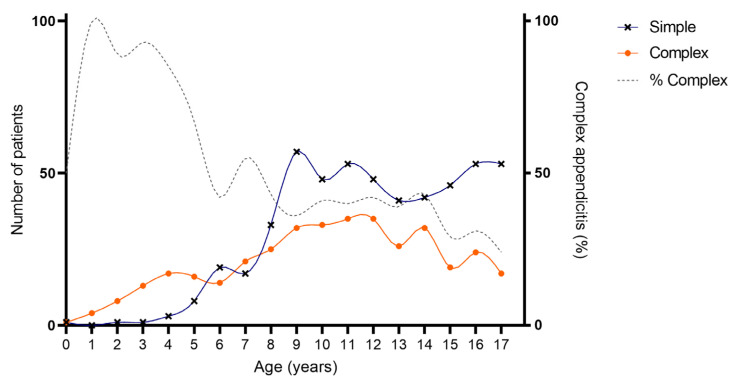
Age of included patients for simple and complex appendicitis. Cumulative number of patients with appendicitis over a time-span of 10 years: simple appendicitis N = 524, complex appendicitis N = 372.

**Figure 3 children-10-01912-f003:**
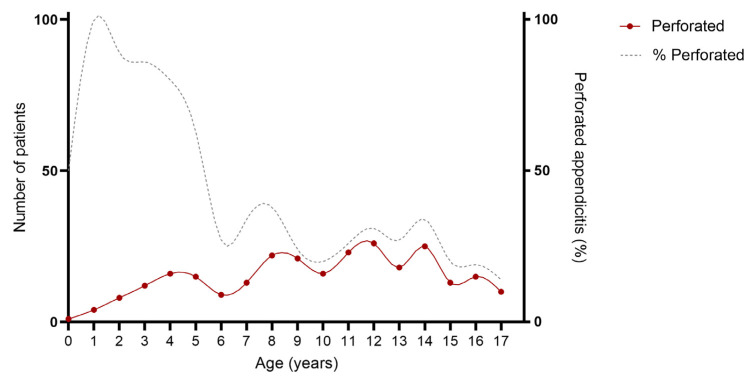
Age of included patients for perforated appendicitis. Cumulative number of patient over a time-span of 10 years: perforated appendicitis N = 267.

**Figure 4 children-10-01912-f004:**
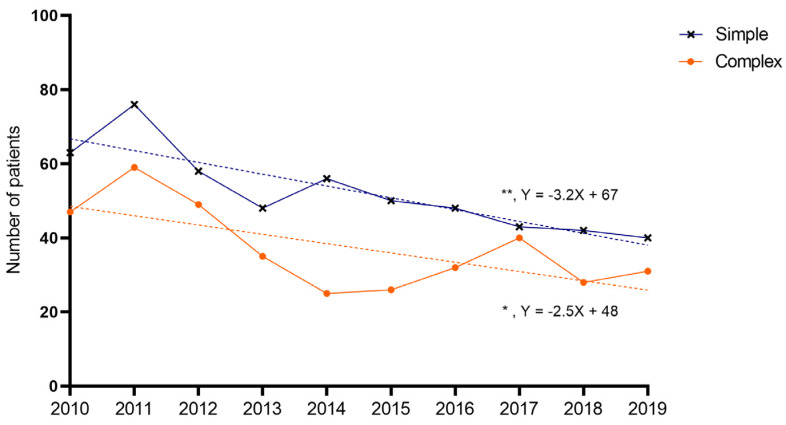
Annual incidence of appendicitis. Dots represent number of patients per year. A significant decrease was noted over the years for simple and complex appendicitis (linear regression analysis, simple slope = −3.188, 95%CI −4.663–−1.713, R^2^ = 0.756, *p* = 0.001; complex slope = −2.509, 95%CI −4.759–−0.259, R^2^ = 0.453, *p* = 0.033), but not for perforated appendicitis.

**Figure 5 children-10-01912-f005:**
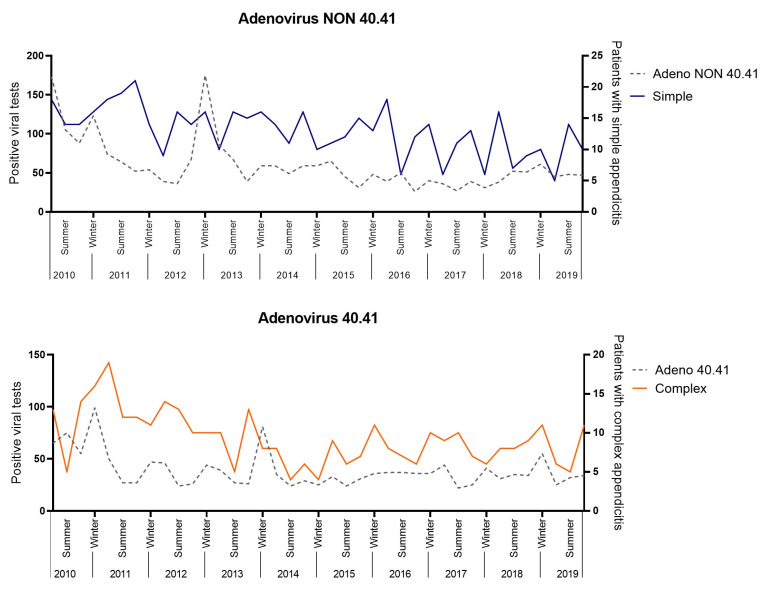
Viral correlation analysis.

**Table 1 children-10-01912-t001:** Baseline characteristics.

	Total N = 896	Simple N = 524	Complex N = 372	*p*-Value
**Hospital**				
Amsterdam UMC	199 (22.2)	79 (15.1)	120 (32.3)	<0.001
NWZ	697 (77.8)	445 (84.9)	252 (67.7)	
**Biological sex**				
Male	519 (57.9)	294 (56.1)	225 (60.5)	0.193
Female	377 (42.1)	230 (43.9)	147 (39.5)	
**Age**	11 [9–14]	12 [9–15]	11 [7–13]	<0.001
**Symptoms**				
Days abdominal pain	1 [1–2]	1 [1–2]	2 [1–3]	<0.001
Nausea	623 (69.5)	357 (68.1)	266 (71.5)	0.303
Vomiting	526 (58.7)	266 (50.8)	260 (69.9)	<0.001
Diarrhea	112 (12.5)	49 (9.4)	63 (16.9)	<0.001
Obstipation	79 (8.8)	27 (5.2)	52 (14.0)	<0.001
Urinary tract symptoms	95 (10.6)	36 (6.9)	59 (15.9)	<0.001
Temperature (°C)	37.5 [36.9–38.1]	37.3 [36.8–37.8]	37.8 [37.1–38.4]	<0.001
**Biochemics**				
Leukocytes × 10^9^/L	15.3 [12.0–18.3]	14.4 [11.4–17.4]	16.8 [13.4–19.8]	<0.001
CRP (mg/dL)	33 [9–88]	16 [6–44]	84 [34–165]	<0.001
**Imaging**				
Ultrasound	877 (97.9)	517 (98.7)	360 (96.8)	1.000
MRI/CT	141 (15.7)	83 (15.8)	58 (15.6)	1.000
Fecolith	203 (22.7)	100 (19.1)	103 (27.7)	0.003

All data is provided as either number with (percentage) or median with [interquartile range]. Statistics were performed with Fisher’s exact test for dichotomous variables and Mann–Whitney U test for continuous variables. Missing data for continuous variables: days of abdominal pain, N = 25; temperature, N = 85; leukocytes, N = 41; C-reactive protein, N = 45.

**Table 2 children-10-01912-t002:** Patients per season.

	Total N = 896	Simple N = 524	Complex N = 372	Perforated N = 267
**Season**				
Spring	235 (26.2)	129 (24.6)	106 (28.5)	70 (26.2)
Summer	204 (22.8)	126 (24.0)	78 (21.0)	59 (22.1)
Autumn	232 (25.9)	139 (26.5)	93 (25.0)	73 (27.3)
Winter	225 (25.1)	130 (24.8)	95 (25.5)	65 (24.3)

All data reported descriptively in number of patients with percentage (%).

**Table 3 children-10-01912-t003:** Patients per month.

	Total N = 896	Simple N = 524	Complex N = 372	Perforated N = 267
**Month**				
January	74 (8.3)	40 (7.6)	34 (9.1)	23 (8.6)
February	71 (7.9)	47 (9.0)	24 (6.5)	17 (6.4)
March	81 (9.0)	47 (9.0)	34 (9.1)	20 (7.5)
April	74 (8.3)	44 (8.4)	30 (8.1)	20 (7.5)
May	80 (8.9)	38 (7.3)	42 (11.3)	30 (11.2)
June	82 (9.2)	52 (9.9)	30 (8.1)	21 (7.9)
July	56 (6.3)	35 (6.7)	21 (5.6)	19 (7.1)
August	66 (7.4)	39 (7.4)	27 (7.3)	19 (7.1)
September	80 (8.9)	53 (10.1)	27 (7.3)	21 (7.9)
October	79 (8.8)	41 (7.8)	38 (10.2)	28 (10.5)
November	73 (8.1)	45 (8.6)	28 (7.5)	24 (9.0)
December	80 (8.9)	43 (8.2)	37 (9.9)	25 (9.4)

## Data Availability

The data presented in this study are available on reasonable request from the corresponding author. With the exception of data on viral pathogens as they were provided by the Dutch Working Group on Clinical Virology from the Dutch Society for Clinical Microbiology (NVMM) and participating laboratories.

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
