# Peer review of "Trends in Simple and Complex Appendicitis in Children and the Potential Correlation to Common Viral Pathogens—A Retrospective Cohort Study between 2010 and 2019 in The Netherlands"

_children, 2023, doi:10.3390/children10121912_

Round 1

Reviewer 1 Report

Comments and Suggestions for Authors

Dear colleagues, thank you for your useful and interesting work! However, I have one major question about the study design. Why, to compare the effect of viruses on the development of appendicitis, were average data on viral persistence in the entire country taken - and not specifically from patients with appendicitis? Unfortunately, this no longer allows for a clear comparison of the influence...

Also note - the difference in the course of appendicitis in children of different age groups is a known fact

I think for the correct perception of the article, it makes sense to rephrase the title, and perhaps change the structure of the document and conclusions

Reviewer 2 Report

Comments and Suggestions for Authors

The authors examined annual, seasonal, and monthly trends in children with simple and complex appendicitis and their correlation with common viral pathogens in the Netherlands. The study was designed as a bicenter retrospective study. This is an interesting topic on which few data have been published. The study is well designed and written. However, several important questions should be addressed before a positive decision is made. 

1. Abstract – The authors state that no relevant seasonal variation was found for either type, nor was there a positive correlation with viral pathogens after Bonferroni correction. Please support this statement with results (n (%) and p-values) 

2. Introduction – Since one of the main conclusions of this study is that of the children aged 0to 5 years, 80% had complex appendicitis, compared with only 38% of those aged 6to 18 years, the authors should add more data on the distribution of appendicitis among age groups. In the introduction, please describe perforation rates regarding age in more details. I would suggest inclusion of following data: 

- The clinical presentation of acute appendicitis for children aged five years or younger is often unusual, and establishing the proper diagnosis is often delayed. Patient age is tied closely to the stage of acute appendicitis, so the youngest patients present with more advanced stages of disease and are at greater risk of perforation. Acute appendicitis should be considered in all cases where a child is having abdominal pain, fever, and diarrhoea, which has been shown to be present significantly more often in children with perforated appendix (doi: 10.3390/diagnostics13132275) 

- Younger children are at significantly greater risk of appendicular perforation. Recent study showed significant increase of perforations regarding age as follow: 100% <1 year; 100% 1-2 years; 83.3% 2-3 years; 71.4% 3-4 years; 78.6% 4-5 years and 47.3% 5 years. (doi: 10.1089/sur.2019.175) 

- In addition, possible causes of acute appendicitis should be stated: The exact cause of acute appendicitis remains unknown in approximately 60% of cases. In the remaining cases, it is caused by lumen obstruction. Obstruction of the lumen is most commonly caused by fecal matter or lymphoid hyperplasia; rarely, appendicitis is caused by tumors, intestinal parasites, or foreign bodies (doi: 10.14744/nci.2022.67984) 

3. IRB Statement–The authors state that the study was approved by the Medical Ethics Committee of UMC Amsterdam. Please indicate the approval number and date of approval. 

4. The authors have used several abbreviations without explaining them. Each abbreviation should be explained in the text when it is first mentioned, e.g., UMC, AMC, WMO, IBD, EDC, NVMM, etc. 

5. Baseline characteristics of the study population are inadequately presented. As indicated in section 2.1, history, physical examination, biochemical and urinalysis, radiological examination, and perioperative course were recorded for each patient. The authors were to include other relevant clinical parameters such as BMI, duration of symptoms, clinical findings (pain, rebound tenderness, nausea/vomiting), laboratory data (WBC, neutrophil count, CRP), and radiological findings and compare them between groups. 

6. Discussion – Paragraph 4.2 I fully agree with the authors that the pandemic period may have influenced the trends of appendicitis in children and should not be included in the analysis. In their commentary, they should mention the results of a meta-analysis regarding the incidence of complicated appendicitis during the COVID-19 pandemic, instead of the results of single centre studies. 

7. Limitations of the study – The authors did not even mention the limitations of the study. There are several limitations in this study. All limitations should be listed at the end of discussion. Please update. 

8. As per MDPI standards a minimum of 30 references is required for such type of the study. Please update.

Comments on the Quality of English Language

Minor editing of English language required.

Reviewer 3 Report

Comments and Suggestions for Authors

The submitted manuscript addresses a conceptually interesting and novel topic. I congratulate the authors for it. Below are my comments:

1. Introduction:

1.1. I am missing some additional references to recent findings regarding the immunological characterization of the cecal appendix, especially those referring to pediatric populations. I have attached relevant references

Omling E, Salö M, Stenström P, Merlo J, Gudjonsdottir J, Rudolfson N, Hagander L. Nationwide paediatric cohort study of a protective association between allergy and complicated appendicitis. Br J Surg. 2021;108(12):1491–1497. doi: 10.1093/bjs/znab326. - DOI PMC PubMed

Carvalho N, Barros A, Coelho HO, Moita CF, Neves-Costa A, Pedroso D, Borges FC, Moita LF, Costa PM. A Th2 cytokine profile in appendicular lavage fluid suggests allergy as a possible etiology for acute appendicitis. Mediators Inflamm. 2019;28(2019):8146257. doi: 10.1155/2019/8146257. - DOI PMC PubMed

1.2. I also believe that the manuscript would benefit from a short introductory paragraph regarding the diagnostic complexity of acute appendicitis (especially in the pediatric population), recent efforts to improve its performance using biomarkers (especially non-invasive), multivariate scores and

intelligence, and the potential link between a viral etiology and new diagnostic tools for this entity (consider some commentary on the changes in the CBC and recent derivative tools, such as the platelet-lymphocyte index). When authors use et al, they should accompany it with a period (et al.). Academic nomenclature.

2. Material and methods

2.1 The major limitation I find in this work is the absence of demonstration of viriasis in all patients by a reliable test (PCR). If I have not misunderstood, the authors correlated national epidemiological data with the chronology of the patients, without a direct link between them and the tests. This presents a great risk of bias, and the n of this study falls far short of what would be expected for an ecological study such as the one in question. This should be discussed in detail.

2.2. A clinical and sociodemographic comparison should be made between inclusions and exclusions to evaluate the possibility of selection bias. If this is not done, I recommend commenting on it in the discussion as a limitation.

2.3. The histopathological classification is not specific enough for this type of work. It is necessary to express how acute phlegmonous appendicitis (polymorphonuclear infiltration up to the muscularis propria), acute gangrenous appendicitis (transmural polymorphonuclear infiltration), etc. were defined.

2.4. "T-test or Mann Whitney U test for continuous variables when appropriate". Explain how normality was evaluated in the distribution of quantitative variables if homogeneity of variances (Levene) was evaluated. The reader should not assume that the authors use parametric or nonparametric tests without explanation.

2.5. “Statistical significance was determined as α < 0.05”. To avoid confusing readers, change to P-value. Specify whether it is one-tailed or two-tailed.

3. Results

3.0. Table 1. Although I agree that the population subgroup of children under 5 years of age has a different clinical presentation from the rest of the pediatric patients, I do not like the presentation of the authors. They do not provide measures of central tendency or dispersion for age, nor do they provide any clinical variable (time of evolution, analytical parameters, etc...). Without these data, it is difficult to understand the population to be analyzed.

3.1. Headings 3.1 and 3.2 should not have a title that translates results, but an aseptic and informative title (example: 3.1. Characterization of clinical presentation according to age).

3.2. With an n of less than 1000, I think it is daring to express the results of section 3.2 with such forcefulness.

3.3. Classifying by seasons and not by months is a limitation. If the change is attributed to an atmospheric or epidemiological issue, we would have to consider at least monthly if not weekly periods, and we would have to stratify by municipalities or regions. Comment as a limitation.

3.4. An R of 0.353 is a weak correlation. If this is added to the type of analysis it constitutes, the results are very questionable. Good Bonferroni correction. The authors need to qualify in detail that this is a hypothetical approach and could be, but is not confirmatory.

Additional comments.

The manuscript would benefit from an English revision by a native speaker. Sometimes unnatural expressions are used or verb tenses are mixed (e.g., "Separate data will be provided for the sub-group of patients with perforated appendicitis. Classification was performed").

The use of an unadjusted simple linear regression model in this case is very limited. Same for Pearson. There are multiple potential confounding variables (temperature, atmospheric pressure, pollution, diet, locoregional geographic distribution...). This article is conceptually interesting but has major limitations that need to be discussed in detail.

Comments on the Quality of English Language

Minor english corrections

Round 2

Reviewer 2 Report

Comments and Suggestions for Authors

The authors have responded appropriately to all reviewer's comments and have significantly improved the manuscript. In my opinion, manuscript can be accepted for publication in its present form.

Comments on the Quality of English Language

Minor editing of English language required.